# Advances and Challenges in CRISPR/Cas-Based Fungal Genome Engineering for Secondary Metabolite Production: A Review

**DOI:** 10.3390/jof9030362

**Published:** 2023-03-15

**Authors:** Duoduo Wang, Shunda Jin, Qianhui Lu, Yupeng Chen

**Affiliations:** 1College of Life Sciences, Zhejiang Normal University, Jinhua 321004, China; huii170026@163.com; 2Zhejiang Provincial Key Laboratory of Biotechnology on Specialty Economic Plants, Zhejiang Normal University, Jinhua 321004, China; 3School of Plant Protection, Hainan University, Haikou 570228, China; jin_shunda@163.com; 4Department of Biochemistry and Molecular Biology, School of Basic Medical Sciences, Southern Medical University, Guangzhou 510515, China

**Keywords:** CRISPR/Cas, genetic engineering, fungi, biosynthetic gene clusters, secondary metabolites

## Abstract

Fungi represent an important source of bioactive secondary metabolites (SMs), which have wide applications in many fields, including medicine, agriculture, human health, and many other industries. The genes involved in SM biosynthesis are usually clustered adjacent to each other into a region known as a biosynthetic gene cluster (BGC). The recent advent of a diversity of genetic and genomic technologies has facilitated the identification of many cryptic or uncharacterized BGCs and their associated SMs. However, there are still many challenges that hamper the broader exploration of industrially important secondary metabolites. The recent advanced CRISPR/Cas system has revolutionized fungal genetic engineering and enabled the discovery of novel bioactive compounds. In this review, we firstly introduce fungal BGCs and their relationships with associated SMs, followed by a brief summary of the conventional strategies for fungal genetic engineering. Next, we introduce a range of state-of-the-art CRISPR/Cas-based tools that have been developed and review recent applications of these methods in fungi for research on the biosynthesis of SMs. Finally, the challenges and limitations of these CRISPR/Cas-based systems are discussed and directions for future research are proposed in order to expand their applications and improve efficiency for fungal genetic engineering.

## 1. Introduction 

Fungi are a major source of secondary metabolites (SMs), also referred to as natural products, and defined as a large diversity of low-molecular-weight organic compounds that are synthesized from simple and inorganic precursors SMs are not directly involved in growth and development; rather, they convey a selective advantage promoting in the survival and fitness of the producing organism [1]. Although a wide range of fungi-derived SMs have already been identified, many SMs remain unknown. So far, approximately 120,000 fungal species have been identified; nevertheless, this number only accounts for less than 8% of the estimated total number of fungal species existing on earth [2]. Furthermore, only a small percentage of SMs have been identified from fungi due to the technical challenges of discovering and identifying novel SMs. 

Fungi can produce a diversity of SMs, including not only beneficial SMs that can be developed into pharmaceutical, agrochemical, and cosmetic products, but also those with negative impacts on humans, plants, livestock, and the environment. For instance, phytotoxins produced by plant pathogenic species can cause many crop diseases, resulting in considerable economic losses and environmental problems [3]. Mycotoxins, a group of toxic compounds that are formed via the metabolism of specific fungi, pose a threat to livestock production and human health [4]. In contrast, many valuable SMs have also been produced from fungi and widely applied in various fields, including the manufacturing industry, agriculture, and medicine. For instance, lovastatin and taxol produced from unique de novo biochemical pathways in filamentous fungi are influential drugs that can treat hypercholesterolemia and cancer [5]. The source species, molecular structures, commercial products, and modes of action of a range of representative fungal SMs have been comprehensively summarized [6]. SMs are synthesized via various pathways using primary metabolites as building blocks, which are categorized into several molecular classes that include polyketides, terpenoids, and non-ribosomal peptides.

The genes responsible for SM biosynthesis in the fungal genome are typically arranged adjacent to each other in the form of biosynthetic gene clusters (BGCs). A fungal biosynthetic gene cluster (BGC) typically contains genes encoding core synthases/synthatases, biosynthetic tailoring enzymes, regulators, and transporters, as well as enzymes related to self-resistance [7]. The number of publicly available fungal genomes has tremendously increased in recent years due to the rapid development of advanced sequencing technologies and genomic tools. This accumulation of annotated genomic information has accelerated the identification of BGCs with the aid of simultaneously developed automated genome mining tools, such as antiSMASH, MIBiG 2.0, and BiG-SCAPE [8,9,10]. By performing bioinformatic analysis of 1037 fungal genomes, Robey et al. [11] found that the number of BGCs encoded by each fungal genome varied greatly between species. Moreover, BGCs vary greatly in size, spanning from a few kb (harboring two genes) to ∼100 kb (containing up to 27 genes) [12,13]. Our understanding of the link between fungal BGCs and their associated SMs is very limited, not only because many well-characterized BGCs are transcriptionally silent, but also because a significant number of BGCs have yet to be explored for their biosynthetic potential [14]. Thus, activating silent BGCs and exploring novel BGCs in the fungal kingdom is an essential precondition for the identification of novel SMs. 

Numerous factors have been shown to regulate the expression of BGCs in fungi, including environmental signals, global regulators, and cluster-specific transcription factors (TFs), as well as epigenetic factors [15]. Crosstalk and interactions between these factors have been observed during the biosynthesis of fungal secondary metabolites. Among these factors, environmental signals and global regulators normally have a regulatory effect on the transcription of multiple BGCs, while cluster-specific regulators/TFs typically regulate only a specific BGC. A number of global regulators involved in BGC regulation have been described, including the velvet complex [16], BrlA [17], laeA [18], and McrA [19]. The expression of some BGCs is specifically controlled by cluster-specific TFs, and the expression levels of these TFs are closely associated with BGC activation. For TFs possessing weak native promoters, promoter replacement or TF overexpression appear to be effective in activating a previously silent BGC [20]. For example, promoter replacement of the cluster-specific transcriptional factor ATEG_06205 in *Aspergillus terreus* resulted in the activation of a polyketide biosynthesis gene cluster, as well as the production of highly pigmented naphthoquinones [21]. In *A. terreus*, overexpressing the pathway-specific transcription factor *tazR* using the Tet-on system activated the taz pathway and induced the production of novel azaphilones [22]. Epigenetic regulation is also critical to gene activity and occurs through various forms, including DNA methylation rewriting, histone modification, small RNA expression, and the modulation of high-order chromatin structures [15]. The reprogramming of the epigenome in fungi is emerging as a promising strategy for altering BGC activity and promoting SM biosynthesis. However, some BGCs are active under certain conditions. In order to identify SMs that are regulated by these active BGCs, knock-out strains are usually generated through gene deletion or disruption, followed by subsequent metabolite profiling. The exploration of the regulatory mechanisms of BGC expression and their connections to SM biosynthesis provides a theoretical basis for the design and evaluation of practical strategies for SM production from fungi. This review firstly summarizes general aspects of fungal secondary metabolism, including the significance of SMs and their encoding genes, followed by a brief discussion of conventional strategies used for genomic engineering in fungi. We then provide a detailed overview of recent advancements in the application of the CRISPR/Cas system for fungal genome engineering and SM production. In addition, the advantages and challenges of various CRISPR/Cas systems are compared. Finally, we point out the challenges and limitations of the currently developed CRISPR/Cas systems, and propose potential solutions and directions for future work to widen the implementation of CRISPR/Cas technology for genome engineering in fungi.

## 2. Conventional Strategies for Fungal Genetic Engineering

Prior to the advent of CRISPR/Cas technology, a diversity of conventional methods have been used to edit fungal genomes and regulate gene expression, including random DNA integration, gene-targeting technology, and RNA technology. Random DNA integration can be created by restriction enzyme-mediated integration (REMI), *Agrobacterium tumefaciens*-mediated transformation (ATMT), and transposon-arrayed gene knockouts (TAGKO) [23]. However, the process of random integration is tedious. Gene-targeting technology is primarily based on homologous recombination (HR), which is widely used for precise gene editing and gene knock-in when a donor DNA template is provided. However, gene-targeted technologies may not be effective in certain fungal species, such as filamentous fungi, due to low rates of HR efficiency [24]. Low HR efficiency in filamentous fungi is due to the requirement for a long, homologous sequence for efficient foreign DNA integration [25,26]. In contrast, HR efficiency is much higher in yeast than in filamentous fungi [24]. On the other hand, many fungal species prefer to use the widely conserved nonhomologous end joining (NHEJ) approach for repairing DNA damage, which, in turn, decreases the HR frequency of gene-targeting [24]. It has been reported that disrupting the NHEJ pathway by suppressing key molecules involved in NHEJ, such as KU70, KU80, and DNA ligase IV, could improve the HR efficiency and further enhance the frequency of precise genetic modifications in filamentous fungi [27]. These conventional methods have been widely used for producing a diversity of bioactive SMs via modulating BGCs in fungi, particularly model organisms and industrially important strains. However, these tools have shown several major disadvantages, including low efficiency, being time-consuming, and low availability of precise genetic markers [28,29,30]. Additionally, difficulty in transformation and screening, and a lack of a vector system have impeded their application in non-modern fungal strains.

The recent introduction of modern gene-editing technologies, especially the CRISPR/Cas system, has revolutionized high-efficiency genetic engineering in fungi by overcoming the aforementioned constraints, opening a new channel for discovering and producing important SMs. CRISPR stands for Clustered Regularly Interspaced Short Palindromic Repeats, and was originally discovered as an antiviral immune defense system in most archaea and many bacteria [31,32]. According to up-to-date evolutionary classification criteria, CRISPR/Cas systems are classified into class I and class II systems, including six types [33]. CRISPR/Cas 9 in type II from the Class 2 CRISPR/Cas system has been extensively explored and exploited for gene editing, which is composed of endonuclease Cas9, CRISPR-derived RNA (crRNA), and trans-activating CRISPR RNA (tracer RNA) [34]. Cas9 is guided by a hybrid of crRNA-tracer RNA to the target DNA sequence and cuts the double-stranded DNA to form a double-strand break (DSB) [35]. The DSB can be subsequently repaired through several cellular DNA repair mechanisms [36]. The error-free, template-dependent HR and the error-prone, template-independent classic NHEJ represent two major pathways that cells use for DNA repair [37,38,39]. Additional pathways include microhomology-mediated end joining (MMEJ) and single-strand annealing (SSA), both of which are error-prone [24]. During the process of DSB repair, random mutations can be induced at the target site via NHEJ, or precise genome editing can be achieved through HR when a DNA donor template is induced. To make the application of the CRISPR/Cas9 system more convenient, dual-tracrRNA:crRNA was engineered as a single RNA chimera, which was also able to direct site-specific DNA cleavage by Cas9 [31]. To date, a variety of CRISPR-based approaches have been established in fungi and successfully applied for SM pathway regulation via genome editing, transcriptional regulation, or epigenetic modification. The structure and mechanisms of various CRISPR/Cas tools are illustrated in Figure 1 and a comparison of these approaches for fungal genetic engineering is listed in Table 1. These efficient, versatile, and programmable CRISPR/Cas systems have shown considerable potential for fungal genetic engineering and novel bioactive substances production.

In addition to CRISPR, gene manipulation in fungi has also been achieved using transcription activator-like effector (TALE) nuclease (TALEN), which consists of a DNA-binding domain and a DNA-cleavage domain originating from TALE and FokI endonucleases, respectively [39]. TALEN and TALE transcription factor fusion protein techniques were firstly used for gene disruption and transcriptional regulation in the filamentous fungus *Trichoderma reesei*, which has been recognized as an excellent cell factory for producing heterologous proteins [40]. More recently, a gene disruption method using TALENs coupled with exonuclease overexpression has been developed for efficient gene editing in *Rhizopus oryzae* [41]. Despite the fact that the construction of TALE repeats is laborious, TALEN has several advantages over CRISPR, such as having broader range of target sites, lower off-target effects, as well as higher genome editing efficiency in heterochromatin regions [42,43].

## 3. Application of CRISPR/Cas Systems in Fungal Genetic Engineering

### 3.1. Classification of CRISPR/Cas Systems

Classification of CRISPR/Cas systems has been reassessed and updated several times due to the increasing diversity of identified CRISPR/Cas systems. Two representative CRISPR/Cas classifications were described in *Nature Reviews Microbiology* in 2011 and 2015 [44,45]. Recent advances in the study of CRISPR/Cas systems that have occurred since 2015 challenge previous classifications and promote the proposal of the latest classification. Based on the new classification, CRISPR/Cas systems are classified into two classes (Class I and Class II), including 6 types and 33 subtypes [33]. In comparison with the 2015 classification system, which includes 5 types and 16 subtypes, the new Class I CRISPR/Cas system includes 3 types (type I, III, and IV) and 16 subtypes, while the new class II CRISPR/Cas system, which has undergone a drastic expansion, includes 3 types (type II, V, and VI) and 17 subtypes [33]. The most widely used CRISPR/Cas9-based gene-editing tools were developed from the type II-A CRISPR/Cas system from *Streptococcus pyogenes.* The type II CRISPR/Cas9 system contains four cas genes, including *cas1*, *cas2*, *cas9*, and *csn2* as a single operon and the CRISPR array. The CRISPR array is transcribed into two parts, including one long precursor CRISPR RNA (pre-crRNA), which is then cleaved into individual CRISPR RNAs (crRNAs), and one small trans-activating CRISPR RNA (tracrRNA), which is complementary to the CRISPR repeat sequence. Guided by small crRNA, Cas9 alone performs interference by introducing DSBs at target sites [46]. A 5′-NGG-3′ protospacer-adjacent motif (PAM) sequence is required for Cas9 cleavage, which is absent from the CRISPR array and thus prevents self-DNA cleavage [46]. The rapid development of CRISPR/Cas systems enables us to develop a diversity of CRISPR-based tools that can be used for fungal genetic modifications with much greater efficiency than traditional strategies, such as homologous activation and heterologous expression. Advances and challenges in CRISPR/Cas-based fungal genome engineering for secondary metabolite production are discussed in the following subsections.

### 3.2. DNA-Based CRISPR/Cas9 System

Previous studies using the CRISPR/Cas9 system for fungi genome editing primarily rely on DNA-based strategies for delivering Cas9 and sgRNA expression cassettes into the nucleus. Cas9/sgRNA expression cassettes are expressed when they are integrated into the fungal genome, followed by the formation of the Cas9 ribonucleoproteins (RNPs) complex in vivo. DNA-based CRISPR/Cas9 gene-editing systems require the construction of species-specific DNA expression vectors and a well-established fungal strain for transformation, which has been widely used in fungi for producing a diversity of SMs [47]. For example, an in vivo expression of the CRISPR/Cas system, using the nuclear localization signal (NLS) from histone H2B for Cas9 delivery, combined with the promoters of the U6 small nuclear RNA or 5S rRNA for sgRNA expression, was established in the filamentous fungus *Fusarium fujikuroi*. This system was applied successfully in *F. fujikuroi* for rewriting the gibberellic acid (GA) metabolic pathways and changing the GA product profile [48]. Knocking out esterase-encoding genes *IAH1* and *TIP1* by CRISPR/Cas9 in *Saccharomyces cerevisiae* increased the abundance of esters and promoted aroma formation [49]. Blocking the competing metabolic pathways by knocking out the rate-limiting enzymes for fatty acid synthesis and sterol synthesis in filamentous fungi using CRISPR/Cas9 could significantly improve the yield of globally marketed drugs, including lovastatin and taxol, which were proven to be more efficient and powerful than traditional methods [5]. CRISPR/Cas9 knockout technology was also used in *Alternaria alternata* to unravel the biosynthetic pathway for the biosynthesis of alternariol and its derivatives, which are common SMs that act as pathogenicity factors [50]. Thus, elucidating the biosynthetic and metabolic pathways of fungal SMs with the aid of CRISPR/Cas systems can be effectively used to develop practical strategies to boost the yield of valuable bioactive SMs and reduce the production of toxic SMs.

In a more recent study carried out on *Glarea lozoyensis*, replacing a proline hydroxylase gene *gloF* with another gene *ap-htyE* via a CRISPR/Cas9-expression plasmid reduced the level of pneumocandin C0 and increased the production of the antifungal drug caspofungin–pneumocandin B0 [51]. Moreover, CRISPR/Cas9 is efficient in constructing genetically modified fungal strains that could function as platform strains for novel SM production. For example, a SM-deficient strain of *Penicillium rubens,* which was created through the consecutive deletion of four BGCs using CRISPR/Cas9, acted as a platform for the integration of the heterologous Calbistrin gene cluster, generating a novel strain that produced a high level of decumbenones, as well as a clean SM profile due to reduced interference from endogenous SMs [52]. Multiple DSBs would limit the survival of fungi, restricting the use of the CRISPR/Cas system for multi-gene editing [53]. To overcome this problem, donor DNA can be introduced into fungi together with Cas9/sgRNAs [54]. This strategy has been reported in a recent study where an improved CRISPR/Cas9 system with a DNA repair template was used for the deletion of a number of sorbicillinoid biosynthetic genes in *Acremonium chrysogenum*, which was highly effective in reducing the level of sorbicillinoids expressed and increasing the production of cephalosporin C [53]. More recently, CRISPR/Cas9-meidated deletion of *Acaxl2*, a key gene regulating arthrospore formation, in industrial *A. chrysogenum* FC3-5-23 resulted in significantly enhanced cephalosporin C production, revealing a close link between mycelial morphology and cephalosporin C production [55].

The majority of fungal species preferentially employ NHEJ over homology-directed repair (HDR) to repair DSB [56]. To increase the efficiency of HDR for precise genetic modifications, mutant fungal strains with an impaired NHEJ pathway have been constructed using DNA-based CRISPR/Cas systems. For example, the mutant strain ku70Δku80Δ, deficient in KU70 and KU80, was generated for the first time in *Scheffersomyces stipitis* by a CRISPR-mediated knockout method, which showed significantly improved HDR-based genome editing efficiency when compared with the parental strain without the KU deletion [57]. Similarly, rewriting the hypocrellin pathway using CRISPR/Cas9 in a NHEJ-deficient mutant strain of *Shiraia bambusicola* resulted in an increase in hypocrellin production by about 12-fold compared with that of the wild-type strain [58].

### 3.3. CRISPR/Cas9 Ribonucleoproteins (RNPs)

In some fungal species or strains, Cas endonuclease and sgRNA DNA expression cassettes cannot be efficiently expressed. An alternative method of introducing the Cas/sgRNA complex into the fungal nucleus can be achieved by transforming in vitro pre-assembled RNP. The RNP-based CRISPR system is superior to DNA-based CRISPR systems as the RNP-based system avoids strain construction and can be used across different species/strains. A system using in vitro-assembled Cas9 RNP coupled with microhomology repair templates was established and showed a greater gene-targeting efficiency across different genetic backgrounds of *Aspergillus fumigatus* compared with classical-gene replacement systems [59]. The application of this RNP-based system for *A. fumigatus* gene editing provided a simple and universal way to tackle the problem of virulence and antifungal drug resistance in multiple clinical isolates of this strain. An in vitro CRISPR/Cas9 system was established in wild-type *Aspergillus wentii* to delete a negative transcriptional regulator, *mcrA*, which is a master regulator of SM clusters, resulting in the enhanced production of a range of new SMs due to the activation of a polyketide synthase (PKS), BGC [19,60]. RNP complexes of modified Cas9 nuclease and pairs of single guide RNAs were used in *Epichloë* species to eliminate the entire ergot alkaloid biosynthesis cluster, which avoided the production of SMs that are toxic to livestock [61]. This RNP system opens the door to non-transgenic manipulations of a wide range of endophytes and facilitates the generation of mutant strains without toxin genes for forage cultivar improvement. CRISPR RNP-based tools have also been used for precise genetic manipulations in many economically important plant pathogens. In the rice blast fungus *Magnaporthe oryzae*, CRISPR RNP-based tools have been developed for specific base-pair editing, gene replacement, and multiple-gene editing with high precision and speed [62].

### 3.4. A Combination of In Vitro and In Vivo Expression of Cas/sgRNA Complex

For fungi without suitable promoters to express sgRNAs, in vitro-synthesized gRNA can be delivered directly into fungal cells for gene targeting. This not only solves the problem of a lack of sgRNA promoters, but also avoids the time-consuming construction of sgRNA expression cassettes. Liu et al. [63] optimized a CRISPR-based system in the filamentous fungus *T. reesei* through the in vivo expression of a specific codon-optimized Cas9 and in vitro transcription of sgRNA for both site-specific mutagenesis and HR-mediated gene replacement. In *Nodulisporium*, the efficiency of a CRISPR/Cas9-based gene disruption was observed to be very low when sgRNA expression was driven by a U6 promoter. However, the mutagenesis frequency was significantly improved when an in vitro-synthesized sgRNA and a linear marker gene cassette were co-transformed into the strain [64]. This indicates that strategies in the delivery of CRISPR/Cas components may influence gene editing efficiency. In *Aspergillus niger*, a combined CRISPR system involving in vitro-synthesized sgRNA and in vivo-expressed Cas9 plasmid was adopted for galactaric acid production by disrupting genes involved in the catabolism of galactaric acid, which resulted in significantly higher frequency of gene deletion than other deletion methods [65].

### 3.5. CRISPR/Cas12a-Based Gene Editing

In CRISPR/Cas systems, a specific PAM sequence is required for sgRNA-guided DNA recognition and strict cleavage of the target site by the CRISPR nuclease. The PAM sequence required for the commonly used *S. pyogenes* Cas9 (SpCas9) is 5′-NGG-3 ′ [66]. The requirement for a specific PAM at the target site limits the use of CRISPR/Cas9-based gene editing. Using Cas nucleases that recognize a broad range of PAM sequences can expand the target scope and enhance the flexibility of CRISPR-based genetic engineering. The class II system has prevalently been developed for molecular biology owing to its simplicity, in which SpCas9 from *S. pyrogenes,* assigned to type II, and Cas12a (Cas12a) from *Francisella novicida*, *Acidaminococcus* sp., or *Lachnospiraceae bacterium* (i.e., FnCas12a, AsCas12a, and LbCas12a), assigned to type V, are deployed for genetic engineering in fungi [67]. Cas12a, also known as Cpf1, differs from Cas9 in the specificity of the required PAM sequence and DNA cleavage pattern. Cas12a recognizes the 5′-NTN-3′ consensus PAM adjacent to the 5′ end of the displaced strand of the protospacer, with a preference for 5′-TTN-3′ over 5′NTN (where N is not T). Cas12a contains only one RuvC domain, which cleaves both DNA strands at different locations, forming a staggered double-strand break [68]. In contrast to the type II CRISPR system, the Cas12a-associated CRISPR array is processed into a short, mature crRNA of 42–44 nt in length without tracrRNA, which begins with 19 nt of the direct repeat, followed by 23-25 nt of the spacer sequence [68]. The application of the CRISPR/Cas12a system for fungal genetic modifications has been reported in a range of industrial strains, such as *Aspergillus nidulans* [69], *A. aculeatus* [70], *A. oryzae* and *A. sojae* [71]. A multiplexing CRISPR/Cas12a system using a single multi-CRISPR/Cas12a plasmid was able to generate deletions in up to four genes in *Ashbya gossypii* [72]. Moreover, different Cas nucleases have been compared for their efficiency of single- and multiplex-gene targeting in fungi. Kwon et al. [73] conducted the first comprehensive evaluation of different CRISPR approaches with respect to their applicability, scalability, and targeting efficiencies in *Thermothelomyces thermophilus*. Specifically, the gene-editing rates were compared between three different CRISPR nucleases (SpCas9, FnCas12a, and AsCas12a) for single- and multiplex-gene targeting with plasmid-based or RNP-based delivery methods [73]. The results suggest that the gene editing efficiency is affected by the Cas nuclease and the target locus. A very recent study using sanger and nanopore sequencing analysis demonstrated that Cas12a-based ribonucleoprotein (RNP) could induce a spectrum of DNA mutations, ranging from small INDELs to large deletions and insertions, in the genome of *M. oryzae*, which suggests the involvement of multiple DNA-repair pathways in repairing the double-strand staggered break caused by Cas12a [74,75]. Interestingly, the biased DNA variations observed in this study suggested a hierarchy for DNA repair pathway choice, which might be mediated by the epigenome and has significant implications for genome engineering and evolution [24,74]. 

### 3.6. CRISPR/Cas-Mediated Transcriptional Regulation

Many fungal BGCs remain silent or lowly expressed due to tight regulatory control. Strategies to activate BGCs include promoter replacement, TF overexpression, modulation of global regulators, and heterologous expression [20], along with the most recently developed CRISPR-based gene activation. In *Thermomyces dupontii*, a silent PKS-nonribosomal peptide synthase (PKS-NRPS) biosynthetic gene was activated via CRISPR/Cas9-mediated promoter knock-in [76]. Multiple BGCs can also be activated via CRISPR/Cas9-mediated promoter replacement. Kang et al. [77] reported the first application of the CRISPR/Cas9 system for multiplex promoter engineering in order to activate a BGC that was previously silent due to a weak native promoter. In this study, a single-marker multiplexed CRISPR/Cas9 and transformation-associated recombination (TAR) (known as mCRISTAR) were developed and successfully used for the simultaneous replacement of multiple native promoters in a SM, BGC [77]. Despite the fact that the mCRISTAR method induced gene expression via simultaneous multiplexed incorporation of promoters upstream of the BGC, a single CRISPR array containing multiple target sites in the mCRISTAR system impacted DNA synthesis and reduced its flexibility for targeting different combinations of genes. In order to overcome these limitations, multiple plasmids-based CRISPR/Cas9 and TAR (mpCRISTAR) has recently been developed for the multiplexed refactoring of BGCs. Compared with mCRISTAR, this method not only significantly improved the multiplexing capacity of promoter engineering by allowing diverse combinations of differentially refactored BGC constructs, but was also more cost-effective [78].

The transcriptional activation of silent BGCs can also be achieved via CRISPR activation (CRISPRa), in which a deactivated Cas (dCas) is fused to trans-activating effectors [79]. CRISPRa has been used to modulate the expression of genes in fungal BGCs for accelerating bioactive SM discovery. A suite of CRISPRa systems, including CRISPR/dLbCas12a-VPR and CRISPR/dSpCas9-VPR, were developed and assessed for their efficiencies of transcriptional activation in the filamentous fungus *A. nidulans* [80]. The results demonstrated that dCas12a worked better for multigene activation than dCas9, and the use of CRISPR/dLbCas12a-VPR for activating the native nonribosomal peptide synthetase-like (NRPS-like) gene *mica* enhanced the production of microperfuranone. In *P. rubens*, dCas9-VPR, together with an sgRNA module, were introduced into a non-integrative AMA1 vector to generate a genome-editing-free CRISPRa system, and this system was able to activate the cryptic macrophorin BGC [81]. Interestingly, it has been suggested that transcription activation could be improved by fusing dCas9 to multiple activator domains (Román et al., 2019). In contrast, transcriptional repression can be achieved via CRISPR interference (CRISPRi), where deactivated Cas9 (dCas9) is fused to repressors [79]. Using a CRISPRi platform, in which dCas9 was fused to a repressor domain, promoter regions of *Candida albicans* were successfully repressed and the intensity of transcriptional repression depended on the position where the CRISPR complex was targeted in the promoter region [82,83]. Another frequently used method to activate silent BGCs is through heterologous expression, which requires the in vitro isolation of the fungal genome and cloning of entire the BGC. The in vitro cloning of a large size of DNA has remained a key challenge of this technique. A recent CRISPR/Cas9 system was reported, for the first time, for capturing entire groups of BGCs in filamentous fungi [84]. In this system, the genomic DNA extracted from fungi was cleaved by RNA-guided Cas9 endonuclease in vitro, and in combination with in vivo yeast assembly, the entire BGCs were inserted into vectors for heterogeneous expression.

### 3.7. CRISPR/Cas-Mediated Epigenetic Editing

Epigenetic regulation plays a critical role in gene expression, as it affects the readability and accessibility of genes to TFs and is determined by environmental factors and epigenetic markers, including DNA methylation, histone modifications, chromatin remodeling, and microRNA (miRNA) [15]. Global epigenetic changes can be induced by environmental factors or genetic modifications of global epigenetic regulators, while the alteration of epigenetic patterns within a specific locus requires the remodeling of local epigenetic markers. An increasing number of studies have revealed a close correlation between epigenetic changes and SM metabolism [85]. For example, chemicals including DNA methyltransferase inhibition and histone deacetylase inhibition have been used to modulate the epigenetic landscape in *Aspergillus* spp., which has caused significant changes in SM profiles [86]. Conventional methods can also be used to manipulate epigenetic remodelers for epigenome rewriting. Nevertheless, the CRISPR/Cas technology has greatly accelerated the advancement of epigenetic editing in bacteria [87], mammalian cells [88], plants [89], and fungi [90].

In *A. niger*, the deposition of histone marks surrounding a range of secondary metabolic genes (*breF, fuml,* or *fwnA*) was accomplished via a CRISPR/dCas9-mediated epigenetic modification system in which dCas9 was fused to different epigenetic regulators, including histone acetyltransferase and histone deacetylase [91], and in each case, target genes were either activated or repressed as expected. More recently, a histone deacetylase encoding gene *rpd3* was deleted in a marine-derived fungus using CRISPR/Cas9, leading to the activation of a series of novel compounds [92]. All these studies suggest that CRISPR-based epigenomic editing shows great potential for understanding and controlling SM metabolism in fungi. It is assumed that, in addition to Cas9, other Cas variants, such as Cas12a, can be used for fungi epigenome editing, which would expand the scope of potential target sites.

### 3.8. CRISPR/Cas9-Based Marker-Free Gene Editing System

Another major limitation that hampers genetic engineering in fungi is the lack of a sufficient number of selection markers. To overcome this challenge, an AMA1-based plasmid, which harbors the AMA1 sequence and other necessary elements, can be used for marker-free genetic modifications. AMA1 was initially discovered in *A. nidulans* and was found to remain in a free form following transformation, instead of being integrated into the fungal chromosome [93]. Plasmid harboring of the AMA1 sequence is usually transformed with high efficiency and replicates autonomously, independent of the fungal genome. The AMA plasmid can easily be recycled after several rounds of subculturing under nonselective conditions, allowing the reuse of the dominant selection marker(s) during transformation [94]. CRISPR/Cas9-based approaches involving an autonomously replicating AMA1-plasmid have been successfully established for the editing of single or multiple genes in industrial strains of *Aspergillus niger* [95], the edible fungus *Cordyceps militaris* [96], *A. terreus* [97], and *A. oryzae* [98]. For instance, an AMA1-based CRISPR/Cas9 genome-editing system was used in *Paecilomyces variotii* and *Penicillium roquefortii* for creating melanin-deficient strains by knocking out the associated *PKS* genes and investigating the effect of melanin on the heat and UV-C radiation resistance of conidia from these food-associated fungi [99].

In addition, AMA1-based genome editing vectors bearing codon-optimized Cas12a expression cassettes were used for the marker-free mutagenesis of the *AowA* and *sC* genes in *A. oryzae* and the *AswA* gene in *A. sojae* [100]. Apart from the AMA1 plasmid, CRISPR-associated marker-free editing tools involving a telomere vector and its improved version have been developed for knocking out single or multiple genes involved in the biosynthesis of phytotoxic compounds in *Botrytis cinerea* [101,102]. Telomere-based plasmids have the ability to replicate autonomously as centromere-free minichromosomes and can be eliminated without selection pressure, which opens a new door to highly efficient, marker-free gene editing in fungi. Occasionally, the constitutive expression of Cas9 in a self-replicating plasmid has a negative effect on fungal cell growth and metabolism [103]. To circumvent this problem, an in vitro-expressed Cas9 protein can be used to replace the in vivo Cas9 expression cassette. For example, in the citric acid-hyperproducer strain *Aspergillus tubingensis* WU-2223L, a CRISPR/Cas9-based marker-free gene replacement system involving in vitro-expressed Cas9 and a DNA fragment encoding sgRNAs that target both the gene of interest and marker gene was constructed for marker-free gene replacement [104]. Using endogenous genes instead of selection marker genes as screening markers is an alternative approach to solving the issue of limited selection markers. The most recent example of using endogenous genes as a screening marker was reported in a *Monascus spp,* and the resistance selection genes used for genetic engineering in this organism are limited. In this study, a markerless system using mutant strains in which the endogenous gene *mrpyrG* had been deleted was developed for multi-gene modification [105].

## 4. Current Limitations and Future Prospects of CRISPR/Cas-Mediated Fungi Genome Engineering

CRISPR/Cas-based approaches have proven to be effective for gene editing and regulation in many fungal species. An overview of the application of CRISPR/Cas technology in fungi is illustrated in Figure 2. However, several major challenges and limitations still impede their application. One major issue to be considered is the gene-editing efficiency, which is determined by multiple factors, including the Cas enzyme kinetics, sgRNA design, gene copy number, repair template, editing mechanism, and more. For example, current sgRNA design tools usually do not take into account sgRNA’s features, such as its secondary structure, which is supposed to impact the efficiency of CRISPR/Cas-based gene targeting. Thus, more sophisticated computational predictive models that can evaluate sgRNA secondary structures are needed for sgRNA design. The efficiency of CRISPR-based gene modification is affected by the surrounding genetic context, such as the position of the target site, chromatin accessibility, the nucleosome, and transcription factor occupancy of the target site. For example, heterochromatic regions might affect the deployment of a Cas protein or Cas–effector complex to the target site. Therefore, developing new tools to predict the architecture and compactness of the chromatin surrounding the targeted region will help to design optimal sgRNA. Additionally, optimal sgRNA design can be achieved through high-throughput analyses of sgRNA’s sensitivity at different genomic loci. This strategy was reported in a previous study in which multiple target loci using gRNA libraries were assessed by a high-throughput CRISPR-based approach involving in vivo-expressed Cas9 and in vitro-synthesized gRNAs, which avoided the tedious sub-cloning of sgRNA expression cassettes [106].

The repair of CRISPR/Cas-induced DSB by NHEJ has been thought to create small indels (insertions and deletions). However, unexpected on-target mutagenesis was detected using long-read sequencing and long-range PCR genotyping in several recent studies. For example, large deletions and more complex genomic rearrangements were induced by a single-guide RNA/Cas9 at the targeted sites in mouse embryonic stem cells, mouse hematopoietic progenitors, and a human differentiated cell line [108]. In *Sclerotinia sclerotiorum,* plasmid insertions were introduced at CRISPR/Cas9-induced DSBs through NHEJ for DNA repair [109]. In a more recent study in *M. oryzae*, large insertions, large deletions, and deletion plus insertion events were all detected at CRISPR/Cas12a-induced DSBs [74]. Extensive and complex on-target effects suggest that CRISPR/Cas-induced DSBs might be repaired through complex mechanisms involving multiple endogenous DNA repair pathways and extensive crosstalk between different pathways. How multiple DNA repair pathways repair DSB synergistically and what determines the choice of DNA repair pathways need further investigation.

Unexpected off-target effects caused by gene editing is another major limitation that remains be addressed. Several approaches have been reported to reduce off-target effects in various organisms. For instance, a series of progressively shorter sgRNAs for targeted gene editing have been tested in human cells, and truncated RNA (tru-RNA) with 17 or 18 nucleotides showed improved specificities without compromising on target activity [110]. This result might be explained as the truncated RNA-guided nuclease complex being more sensitive to mismatches compared with full-length sgRNA with 20 nucleotides. Interestingly, tru-RNAs with even shorter and longer complementarity lengths (other than 17 or 18 nucleotides) either failed to show activities or showed substantially decreased activities, results that need further investigation [110]. It has been suggested that the use of paired Cas9 nickases with two sgRNAs targeting opposite DNA strands could generate two single-strand breaks (SSBs) or nicks on opposite DNA strands, which avoids or minimizes off-target effects without sacrificing genome-editing efficiency [111]. Reduced off-target effects observed from the use of Cas nickases are probably due to a lower frequency of indels induced by the Cas nickase compared with those induced by the Cas nuclease at off-target sites, providing a new clue for minimizing off-target effects. Additionally, numerous studies have indicated that off-target effects in CRISPR-mediated gene modifications were reduced by using engineered Cas9 variants, including the high-fidelity variant SpCas9-HF1 [112], enhanced specificity (by eSpCas9) [113], and hyper accuracy (by HypaCas9 [114], Cas9-NG [115], and xCas9 [100,116,117]). More recently, the newly engineered SpCas9 variant SpRY nearly eliminated the PAM requirement for SpCas9 in human cells, and it also eliminated nearly all detectable off-target effects [118]. The use of engineered Cas variants can be a powerful and effective way to address the issue of off-target effects. Meanwhile, these Cas variants greatly broaden the PAM compatibility and address the limitation of the PAM requirement for DNA-targeting CRISPR enzymes. Several engineered Cas9 variants, with altered PAM specificities, have been applied in fungi for gene modifications. For example, SpCas9-NG was applied to *S. cerevisiae* for multiplex genome disruption and single-nucleotide conversion [119]. Furthermore, a range of Cas9 variants, including Cas9-VQR, Cas9-VRER, xCas9, and SpCas9-NG, were used for high-precision nucleotide editing in yeast [120]. Future work involving the engineering of novel Cas variants and investigating if these Cas variants could function effectively in fungi for other CRISPR-mediated applications is needed.

Genetic engineering technologies are applied to reveal if specific genes or clusters are involved in the biosynthesis of certain SMs. However, the underlying mechanisms of fungal secondary metabolism and its regulatory network still remain unclear. This might require the integrated analysis of multidisciplinary data involving modern molecular biology, bioinformatics, and omics. In addition, the application of synthetic biology tools makes it possible to construct genetic circuits using various modulars in a highly efficient and controlled manner [121]. With the rapid development in the field of synthetic biology, high-throughput fungal genome manipulations are becoming more feasible. So far, genetic transformation systems are only available for model strains and some industrial stains [122,123]. Thus, establishing and optimizing efficient genetic transformation systems in more fungal strains is necessary for expanding the field of CRISPR-based fungal genetic engineering.

CRISPR/Cas systems can also be used for RNA editing. Recently, a number of CRISPR/Cas systems with RNA-targeting activity have been identified, including type II (Cas9), type III (Cmr/Csm), and type VI (Cas13). Novel CRISPR-based RNA-targeting tools that are developed from Type II, III, and VI systems have already been harnessed for endogenous RNA knockdown, site-specific RNA editing, and RNA tracking in many organisms, including fungi [124,125,126,127,128,129,130]. In yeast, a range of Cas 13 proteins, including Cas13a from *Leptotrichia shahii* (LshCas13a), Cas13a from *Leptotrichia wadei* (LwaCas13a), Cas13d from *Ruminococcus flavefaciens* (RfxCas13d), and Nme1Cas9 from *Neisseria meningitidis*, have been used for gene knockdown via triggering mRNA degradation [124,130]. Moreover, an engineered RNA editing system consisting of dCas13a fused with the catalytic domain of a human adenosine deaminase acting on RNA type 2 (hADAR2d) and a crRNA/pRNA construct was applied in fission yeast for precise base editing [130]. Compared with CRISPR-based DNA-targeting, RNA-targeting systems hold exciting potential to dissect the roles of lethal genes and avoid generating permanent off-target genetic changes. However, the application of CRISPR-based transcriptomic manipulation in fungi is limited to only a few species. Future applications of CRISPR-based RNA-targeting systems along with the already-established CRISPR-based platforms are expected to significantly advance our understanding of fungal secondary metabolism.

## Figures and Tables

**Figure 1 jof-09-00362-f001:**
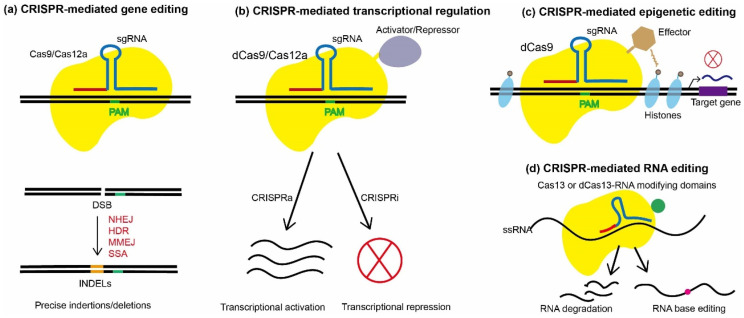
Diagram illustrating the mechanisms of representative CRISPR/Cas-based tools established in fungi. (**a**) CRISPR-mediated gene editing, where Cas9/Cas12a is guided to the target site with the aid of sgRNA and cleaves the target sequence, generating a double-stranded break (DSB). The DSB can be repaired via various repair pathways, including non-homologous end joining (NHEJ), homology-directed repair (HDR), microhomology-mediated end joining (MMEJ), and single-stranded DNA annealing (SSA). Various types of mutations, such as small indels, large deletions, or insertions, can be introduced at the DSB during the repairing process. (**b**) CRISPR-mediated transcriptional regulation, where deactivated Cas9/12a (dCas9/dCas12a) is fused with activation/repression domains, forming a complex that targets the promoter region of the target gene to activate or repress gene expression. CRISPRa: CRISPR-based activation; CRISPRi: inhibition. (**c**) CRISPR-mediated epigenetic editing, where dCas9 is fused with effectors, forming a complex for targeted histone modifications, impacts the expression level of the target gene. (**d**) CRISPR-mediated RNA editing uses Cas13, which has RNA-targeting activity to knockdown gene expression by triggering mRNA degradation, or uses an engineered dCas13-RNA-modifying domains-fusion protein for editing specific nucleotide residues. ssRNA: single-strand RNA; PAM: protospacer adjacent motif.

**Figure 2 jof-09-00362-f002:**
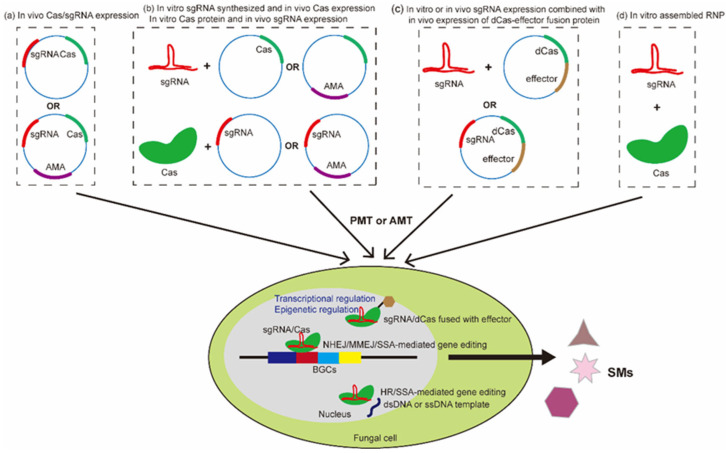
An overview of the application of CRISPR/Cas technology in fungi. The expression of Cas and gRNA can be achieved via several strategies. (**a**) In vivo expression of Cas and gRNA in the form of a plasmid or self-replicating AMA plasmid. (**b**) Cas and gRNA are expressed in a combined way, including in vitro synthesized gRNA and in vivo Cas expression, and in vitro Cas expression and in vivo gRNA expression. (**c**) CRISPR/dCas-based gene regulation and epigenetic editing, achieved via a gRNA/dCas–effector complex which is formed in vitro or in vivo expressed gRNA and a deactivated Cas (dCas)–effector fusion protein. (**d**) Cas and gRNA can both be expressed in vitro to form an RNP complex. An AMA plasmid is used for recycling selection markers. The major methods of genetic transformation for fungi include the protoplast-mediated transformation method and the Agrobacterium-mediated transformation method, *Agrobacteria*-mediated transformation [107]. Guided by gRNA, Cas nuclease induces DNA double-strand breaks (DSBs) at target sites. CRISPR/Cas-based gene editing can be achieved during the process of DSB repair via several DSB repair pathways, including non-homologous end joining (NHEJ), homologous recombination (HR) (dsDNA or ssDNA template is required), microhomology-mediated end joining (MMEJ), and single-strand annealing (SSA) (ssDNA template might be required) [36]. CRISPR/dCas systems are applied for gene expression regulation when dCas is fused to activator/repressor domains, or epigenetic editing when dCas is fused to epigenetic regulators [79]. Cas refers to CRISPR-associated proteins; dsDNA: double-strand DNA; ssDNA: single-strand DNA; BGCs: known as biosynthetic gene clusters, referring to the genomic regions that contain genes encoding enzymes regulating a metabolite biosynthesis pathway; SM: secondary metabolite.

**Table 1 jof-09-00362-t001:** A comparison of several major CRISPR/Cas-based systems for fungal genetic engineering.

CRISPR/Cas-Based Systems	Elements	Mechanisms	Factors Determining Efficiency
CRISPR/Cas-based gene editing system	Cas (Cas9, Cas12a, and other Cas variants); sgRNA	HR-mediated gene editing with a template; NHEJ-mediated gene editing	sgRNA design, gene copy number, PAM specificity, Cas enzyme kinetics, repair template, off-target effect, and genetic context
CRISPR/Cas-mediated gene regulation	dCas9-effector complex; sgRNA	Transcriptional activation or repression via targeting promoters	Numbers and types of effectors, sgRNA design, Cas protein, chromatin structure around targeted promoter region, PAM specificity, and incorporation of RNA aptamers and multimeric peptide arrays
CRISPR/Cas-epigenetic editing	dCas9-epigenetic effectors;sgRNA	Remodeling global or local chromatin context via the deposition of histone marks or genetic modification of epigenetic regulators, thus regulating gene expression as it affects the accessibility of genes to TF	Epigenetic effectors, sgRNA, chromatin context, and PAM specificity
CRISPR/Cas9-based marker-free gene editing	AMA1-based plasmid, telomere vectors, various Cas proteins, and sgRNA	AMA replicates autonomously, independent of the fungal genome, and can be recycled, allowing the reuse of selection markers; telomeres-based plasmids can replicate autonomously as centromere-free mini chromosomes, and be eliminated without selection pressure	sgRNA, chromatin context, PAM specificity, off-target effect, transformation or cotransformation rates, and linearization of pTEL in vivo.
CRISPR/Cas13-based RNA editing	Cas13, dCas13- RNA-modifying domain-fusion protein, and CRISPR RNA (crRNA)/pairing RNAs expression vectors	Cas13 can be used for gene knockdown via its ability to cleave single-strand RNA at sites guided by crRNA; dCas13a-RNA-modifying domains-fusion protein coupling with pairing RNAs can be used for precise RNA base editing	Gene transcript abundance, RNA secondary structure, molar ratio of Cas13/Cas13-RNA modifying domains fusion protein to either crRNA-pRNA or the target transcript, editing activities of RNA-modifying domain, binding ability of Cas13 towards RNA target, and position of editing residue

## Data Availability

No new data were created or analyzed in this study. Data sharing is not applicable to this article.

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
