# Peer review of "Advances and Challenges in CRISPR/Cas-Based Fungal Genome Engineering for Secondary Metabolite Production: A Review"

_jof, 2023, doi:10.3390/jof9030362_

Round 1

Reviewer 1 Report

The review provided the recent fungal biosynthetic gene clusters (BGCs) and their relationships with associated secondary metabolites (SMs) followed by a brief summary of the conventional strategies for fungi genetic engineering. Moreover, the review summarized the recent application of CRISPR/Cas-based tools in fungi for research on the biosynthesis of SMs.

I find the introduction and “conventional strategies for fungi genetic engineering” of the paper quite good, but the other sections are quite poor in the detail examples and clarity for why the researchers chose to CRISPR/Cas-based tools in fungi for the production of SMs.

In general, the manuscript has not covered the application of CRISPR/Cas systems in fungi genetic engineering for the production of SMs. Moreover, the efficiency of CRISPR/Cas systems in the different fungi should be compared in the table. The structure and mechanisms of the CRISPR/Cas system should be covered in the Figure.

Author Response

We thank the reviewer for these useful suggestions. We have addressed the comments. Please see below and revised ms. 

1.      Throughout the text, we have added additional examples to demonstrated in more detail on the application of CRISPR for SM production and clarified why CRISPR tools were chosen for SM production in fungi. Please see line 255-266, line 271-286, line 294-296, line 375-383, line 471-475, line 479-485, line 544-557.

2.      We have added more information in table 1 on the comparison of the efficiency between different CRISPR/Cas systems. The CRISPR/Cas-based strategies mentioned in the table have different purposes, instead of making a direct comparison for the efficiency, we listed all potential factors that might affect the efficiency of each strategy. 

3.      Figure 1 has been added to illustrate the structure and mechanisms of several major CRISPR/Cas systems.

Reviewer 2 Report

Wang et al. present an update on CRISPR/Cas-mediated genome engineering in fungi, with a focus on secondary metabolites. Overall, this review is timely and helpful for the field. However, there are some concerns that need to be addressed before it can be accepted.

------------------------------------------------------------------------------------------------------------------------------------------

Line 2 Please incorporate BGCs and SMs in the title. e.g.,  “Advances and challenges in CRISPR/Cas-based fungal genome engineering for secondary metabolite production: a review”

Line12 Fungi represent.

Line 25 fungal genetic engineering.

Line 31 the definition of SMs is vague, please consider adding more details. e.g., SMs are not directly involved in the normal growth, development, or reproduction of the organism.

Line42-44 please consider citing https://academic.oup.com/femsre/article/37/1/94/558665

Line 64-64 please add references to support your statement.

Line70, https://academic.oup.com/femsre/article/43/6/591/5521207

Is a better reference than [12], same for lines 86 and 307

Line 81, please make sure to italicize the species name.

Line 107 should be Agrobacterium tumefaciens-mediated transformation (ATMT)

Line 111, please consider citing https://academic.oup.com/femsre/article/46/6/fuac035/6638986?searchresult=1 to support your statement about fungal DNA repair.  Same for Lines 114,117, 120, and 142.

Line120, please consider citing the original fungal work https://www.pnas.org/doi/10.1073/pnas.0402780101, instead of the work done in mammalian systems.

Table 1

in the first row, please use Cas12a (also known as Cpf1) throughout the manuscript.

In the last row, considering describe the telomere-based selection system developed in https://journals.plos.org/plospathogens/article?id=10.1371/journal.ppat.1008326 and

https://journals.plos.org/plospathogens/article?id=10.1371/journal.ppat.1010367 please also discuss this system in “3.7 CRISPR/Cas9 based marker-free gene editing system”

Please include Cas13 RNA editing as described in yeast https://academic.oup.com/nar/article/46/15/e90/5025895

Line 207, these two significant studies using RNP in Magnaporthe, should be cited and discussed. Either here or under “3.4. CRISPR/Cas12a-based gene editing”

https://www.nature.com/articles/s41467-022-34736-1

https://star-protocols.cell.com/protocols/1324

Line 225, please consider using Cas12a, rather than Cpf1 throughout the manuscript.

Line 282, it is generally preferred to use abbreviations for species names after the first mention.

Could you review this issue throughout the manuscript?

Line 402, since the authors discussed the off-target effect, please consider also discussing the extensive on-target effect as revealed by several studies including

https://www.nature.com/articles/s41467-022-34736-1

https://journals.asm.org/doi/10.1128/mBio.00567-18

https://www.nature.com/articles/nbt.4192

--------------------------------------------------------------

Author Response

Author response: (The line number was updated for the revised ms, we use updated line numbers for below response)

We thank the reviewer for these useful suggestions. We have addressed the comments according to the advices.

1. In order to incorporate BGCs and SMs into the title, the previous title has been changed into “Advances and challenges in CRISPR/Cas-based fungal genome engineering for secondary metabolite production: a review”.

2. Line 14 “Fungi represents” has been changed into “Fungi represent”.

3. "fungi genetic engineering” has been replaced with “fungal genetic engineering” throughout the text.

4. More details of SMs have been added, see line 33-37.

5. New reference has been cited for Line 46-48.

Woloshuk CP, Shim WB. 2013. Aflatoxins, fumonisins, and trichothecenes: a convergence of knowledge. FEMS Microbiol Rev. 37(1):94-109. doi: 10.1111/1574-6976.12009.

6. Reference has been added for line 70-73.

7. Line 78, 95 and 436. We have replaced previous ref with the below ref suggested.

Collemare J, Seidl MF. 2019. Chromatin-dependent regulation of secondary metabolite biosynthesis in fungi: is the picture complete? FEMS Microbiol Rev. 43(6):591-607. doi: 10.1093/femsre/fuz018.

8. The species name has been italicized

9. Line 116, we have changed the Agrobacterium-mediated transformation (AMT) into “Agrobacterium tumefaciens-mediated transformation (ATMT)”

10. Line 121, 124, 127, and 157. We have cited the suggested paper to support our statement about fungal DNA repair.

Huang J, Cook DE. 2022. The contribution of DNA repair pathways to genome editing and evolution in filamentous pathogens. FEMS Microbiol Rev. 46(6): fuac035. doi: 10.1093/femsre/fuac035.

11. Line 130, the original work has been cited.

Ninomiya Y, Suzuki K, Ishii C, Inoue H. 2004. Highly efficient gene replacements in Neurospora strains deficient for nonhomologous end-joining. Proc Natl Acad Sci U S A. 101(33):12248-53. doi: 10.1073/pnas.0402780101.

12. Telomere-based selection system has been added in the main text as well as the table. Line 479-485.

13. Cas13-based RNA editing has been added in the main text and table 1. Line 607-614.

14. The two significant studies using Cas12a/RNP in Magnaporthe has been cited and discussed under “3.4. CRISPR/Cas12a-based gene editing” Line 375-383

15. Cpf1 has been changed into Cas12a throughout the ms.

16. Abbreviations for species names have been checked throughout the ms.

17. Extensive on-target effect as revealed by the below studies has been discussed in the section 4. Line 544-557.

Reviewer 3 Report

The authors discussed recent data on the use of the CRISPR/Cas system for metabolic engineering of filamentous fungi to increase their biotechnological potential in the production of biologically active secondary metabolites.

Please discuss the results presented in the article (DOI: 10.1007/s00253-017-8263-z).

Since there are other reviews on this topic, such as https://pubs.rsc.org/en/content/articlehtml/2022/np/d2np00055e, please avoid possible plagiarism and provide your perspective on the results of articles already cited in these and similar reviews.

In the text (section 2), also discuss the use of TALEN for editing fungal genomes using publications doi:10.1093/femsle/fnac010, doi: 10.1007/s10295-017-1963-7.

Make a new section to describe CRISPR/Cas systems and discuss in detail the type II-A system from S. pyogenes. You could do a subsection in section 3. And combine the information about the description of the type II-A natural system from other parts of the text (e.g., from section 3.4).

Throughout the text, correct the classification and description of CRISPR/Cas systems, using only recent publications (2018 and later) for citations and correct the text:

Lines 131-134 Please use more recent reviews on this topic and use an up-to-date view of the number of CRISPR/Cas system types.

Line 135 This is the type II that contains Cas9 as a CRISPR effector; other class II types have other Cas effectors, not Cas9.

Lines 235-237 Misunderstanding of modern classification of CRISPR/Cas systems: there are 6 types of systems. Please correct and specify which types belong to which class.

Line 242 - Please specify the PAM consensus for Cpf1. Describe the sgRNA structure in more detail for the CRISPR/Cas type V system.

Please use italicized bacterial and fungal names throughout the text.

Author Response

Author response: (The line number was updated for the revised ms, we use updated line number for below response)

We thank the reviewer for all of the useful suggestions. We have addressed the comments according to the advices.

1. The below article has been discussed in the ms. Section 3.1 and Introduction. Line 52-54, line 255-259

El-Sayed ASA, Abdel-Ghany SE, Ali GS. 2017. Genome editing approaches: manipulating of lovastatin and taxol synthesis of filamentous fungi by CRISPR/Cas9 system. Appl Microbiol Biotechnol. 101(10):3953-3976. doi: 10.1007/s00253-017-8263-z.

2. We have provided our perspective on the cited articles throughout the ms.

3.  Info on TALENs has been added in section 2 and reference cited are listed below: line 168-179.

4. A new section describing CRISPR/Cas systems especially the type II-A system from S. pyogenes has been added, see Section 3.1.

5. Throughout the ms, CRISPR/Cas system classification has been updated, and old refs have been replaced with the recent one. Line 142-144, section 3.1

6. Description on Cas9 has been corrected. Line 145.

7. the modern classification of CRISPR/Cas systems has been corrected Section 3.1

8. the PAM consensus for Cpf1 and the sgRNA structure for the CRISPR/Cas type V system has been described. Line 355-362.

9. Bacterial and fungal names have been italicized throughout the text.

Round 2

Reviewer 1 Report

The revised manuscript should be accepted for publication

Author Response

We have addressed the comments

Reviewer 3 Report

The authors have greatly improved the manuscript by responding to all the reviewers' comments. I have no further comments and can recommend the manuscript for publication in the journal.

Author Response

we have addressed the comments